# Selection on plastic adherence leads to hyper-multicellular strains and incidental virulence in the budding yeast

Luke I Ekdahl[†], Juliana A Salcedo[†], Matthew M Dungan[‡], Despina V Mason, Dulguun Myagmarsuren, Helen A Murphy*

Department of Biology, College of William and Mary, Williamsburg, United States

**Abstract** Many disease-causing microbes are not obligate pathogens; rather, they are environmental microbes taking advantage of an ecological opportunity. The existence of microbes whose life cycle does not require a host and are not normally pathogenic, yet are well-suited to host exploitation, is an evolutionary puzzle. One hypothesis posits that selection in the environment may favor traits that incidentally lead to pathogenicity and virulence, or serve as pre-adaptations for survival in a host. An example of such a trait is surface adherence. To experimentally test the idea of 'accidental virulence', replicate populations of *Saccharomyces cerevisiae* were evolved to attach to a plastic bead for hundreds of generations. Along with plastic adherence, two multicellular phenotypes— biofilm formation and flor formation— increased; another phenotype, pseudohyphal growth, responded to the nutrient limitation. Thus, experimental selection led to the evolution of highly-adherent, hyper-multicellular strains. Wax moth larvae injected with evolved hyper-multicellular strains were significantly more likely to die than those injected with evolved non-multicellular strains. Hence, selection on plastic adherence incidentally led to the evolution of enhanced multicellularity and increased virulence. Our results support the idea that selection for a trait beneficial in the open environment can inadvertently generate opportunistic, 'accidental' pathogens.

*For correspondence:
hamurphy@wm.edu

[†]These authors contributed equally to this work

Present address: [‡]School of Medicine, Vanderbilt University, Nashville, United States

**Competing interest:** The authors declare that no competing interests exist.

## Editor's evaluation

Using experimental evolution and virulence assessment in a model system, this valuable study examines how yeast virulence can coincidentally evolve following selection for plastic adherence. The strength of evidence is solid. The work presents interesting experimental systems, and the findings will be of interest to investigators in the field of experimental evolution and evolution of fungal pathogens.

## Introduction

The study of infectious disease often focuses on pathogenic microbes that either specialize in exploiting animal hosts or act as commensals that switch to pathogenesis when the delicate balance between host and microbe is perturbed. These microbes are presumed to have co-evolved complex adaptations that allow survival and reproduction in and on hosts. However, there exists a broad range of microbial organisms that live in the open environment (i.e. soil, vegetation, aquatic habitats) that are capable of causing disease when the opportunity presents itself (*Brown et al., 2012*). Such microbes also have adaptations that allow host exploitation, but the origin of these adaptations is unclear, as growth and survival in a host is not a required part of the lifecycle (*Casadevall and Pirofski, 2007*). For example, the soil-associated bacteria *Pseudomonas aeruginosa* (*Selezska et al., 2012*) and *Burkholderia cepacia* (*Mahenthiralingam et al., 2008*) can infect the lungs of cystic fibrosis patients.

**eLife digest** Yeast are microscopic fungi that are found on many plants, in the soil and in other environments around the world. But, when given the chance, some yeasts are also good at infecting human and other animals and causing disease.

It has been proposed that some opportunistic microbes may have dual-use traits that evolved for one purpose in their natural environment but also incidentally allow them to infect animals. For example, a toxin that helps the opportunistic microbe compete against neighboring microbes may also weaken an animal. Or the ability of many individual microbe cells to clump together into structures known as biofilms on solid surfaces, or floating mats called flors on liquids, helps them to survive in harsh environments, whether in the soil or in the body of an animal.

To investigate this possibility, Ekdahl, Salcedo et al. examined whether artificially selecting yeast with a specific trait – the ability to stick to plastic beads – in the absence of any host animals would inadvertently also select for yeast that were good at causing disease. This trait was selected because it has not been previously linked to opportunistic yeast infections.

The team grew the yeast for 400 generations in tubes that each contained a plastic bead. At every generation, only yeast that stuck to the plastic bead were transferred to a fresh tube to grow the next generation. The experiments found that, not only did the ability of the yeast to stick to the plastic increase over time, but the yeast also evolved the ability to form biofilms and flors. Furthermore, the sticky yeast killed an insect host known as wax moth larvae more quickly than non-sticky yeast.

Together, these findings demonstrate that when microbes evolve in an environment that is devoid of any host animals, selection can inadvertently favor dual-use traits that also help the yeast to infect animals. Opportunistic yeast infections are of increasing concern in human patients, particularly those with weakened immune systems. Understanding which yeast traits are dual-use will help guide future efforts in combatting yeast and other opportunistic microbes.

The existence of virulence traits in environmentally-derived opportunistic pathogens may be due to selection favoring the traits for other uses in the non-host environment, which challenges the idea that co-evolution is a requirement of microbial pathogenesis and virulence. This hypothesis, first proposed and explored in the bacterial literature, is known as 'coincidental selection' (*Levin and Svanborg Edén, 1990*). Much of the research testing the coincidental selection hypothesis has focused on the biotic environment as a selective pressure, specifically the role of predatory bacteriophages and protists (reviewed in [*Sun et al., 2018*; *Davies et al., 2016*; *Erken et al., 2013*]). The relationship between virulence, bacteria, and their predators is complex (*Brüssow, 2007*). For example, a positive relationship between amoeba predation and virulence was shown in *Escherichia coli* (*Adiba et al., 2010*), while selection by amoeba predation led to a decrease virulence in *P. aeruginosa* (*Leong et al., 2022*). In the same bacterial species, an evolution experiment with both protist and phage predators showed that phage could mitigate the decrease in virulence associated with protist predation (*Friman and Buckling, 2014*). Despite the complexity, it is clear that selection from the biotic environment can strongly influence bacterial virulence.

A hypothesis similar to coincidental selection, known as 'accidental virulence', has been proposed (*Casadevall and Pirofski, 2007*) and explored in parallel in the eukaryotic microbial literature, also with a focus on the biotic environment (reviewed in *Siscar-Lewin et al., 2022*; *Casadevall et al., 2019*). For example, in the fungus *Cryptococcus neoformans*, traits that protect from predatory amoeba also play a role in human infection (*Steenbergen et al., 2001*), and selection by amoeba can increase the prevalence of such traits (*Fu et al., 2021*; *Steenbergen et al., 2003*). Similarly, co-culturing *Paracoccidioides* fungi with amoeba can lead to increased virulence (*Albuquerque et al., 2019*).

In both the bacterial and eukaryotic literature, there has been less focus on the role of selection imposed by the abiotic environment, although temperature is increasingly a consideration, as thermotolerance and halotolerance may make colonization in and on humans more likely (*Casadevall, 2020*; *Garcia-Solache and Casadevall, 2010*). For example, the emerging opportunistic yeast *Candida auris*, which is found in warm and salty coastal wetlands, can cause severe systemic infection (*Arora et al., 2021*). There is also experimental evidence that increased temperature can select for virulence in *P. aeruginosa* (*Friman et al., 2011*).

The dual-use virulence traits that can be under 'coincidental selection' and lead to 'accidental virulence' are numerous and range from toxin production, such as the production of gliotoxin in the soil-dwelling filamentous fungus *Aspergillus fumigatus* (*Hillmann et al., 2015*; *Gupta et al., 2021*), to protective structures, such as capsule formation in *C. neoformans* (*Casadevall et al., 2003*). Another type of trait, which is the focus of the research presented here, is adherence. Adherence is important for many microbial behaviors required for survival (e.g., biofilm formation) (*West et al., 2007*), but can also play a role in pathogenicity and virulence (*Douglas, 2003*; *Hall-Stoodley et al., 2004*). The ability to adhere to and invade tissues, as well as form communities resistant to anti-microbials, can be key to successful pathogenicity. In the soil-associated yeast, *Blastomyces dermatitidis* (*Baumgardner and Laundre, 2001*), which can cause lung infections, the deletion of a single adhesin gene abolishes pathogenesis (*Klein, 2000*).

The function of microbial traits in both the open environment and the host is crucial evidence for the coincidental selection-accidental virulence hypothesis. However, experimental evolution allows the idea to be tested directly; thus far, most evolution experiments have focused on co-culturing opportunistic microbes with predators (e.g., [*Leong et al., 2022*; *Friman and Buckling, 2014*; *Fu et al., 2021*; *Steenbergen et al., 2003*; *Albuquerque et al., 2019*; *Friman et al., 2011*; *Mikonranta et al., 2012*; *Hosseinidoust et al., 2013*]). In the research presented here, rather than altering the biotic or abiotic environment and determining the effect on virulence, we apply direct selection to a specific trait hypothesized to be dual-use in the biomedical model yeast *Saccharomyces cerevisiae*.

Aside from serving as a model for genetics and cell biology, and being found in a myriad of ecological niches around the globe (*Peter et al., 2018*), *S. cerevisiae* is also an opportunistic pathogen capable of infecting immunocompromised individuals, with reports of infections increasing (*Aucott et al., 1990*; *Muñoz et al., 2005*; *Llopis et al., 2014*; *Enache-Angoulvant and Hennequin, 2005*; *Hennequin et al., 2000*; *Pérez-Torrado and Querol, 2015*). As such, it has been a model for fungal pathogenesis-related traits (*Clemons et al., 1994*; *Fraser et al., 2012*; *McCusker et al., 1994a*; *Phadke et al., 2018*). Some environmentally derived strains are capable of adhering to surfaces and expressing associated aggregative phenotypes (*Verstrepen and Klis, 2006*). These range from biofilms on solid and semi-solid agar, to pseudohyphal growth and agar invasion, to floating mats on liquid surfaces (flors). Of these multicellular phenotypes, only invasive and pseudohyphal growth have been linked to pathogenicity in *S. cerevisiae* (*McCusker et al., 1994a*; *Phadke et al., 2018*; *Palecek et al., 2002*), although biofilm formation has been linked to pathogenicity in other fungi (*Fanning and Mitchell, 2012*). Not all strains are capable of expressing these phenotypes. And while some strains express multiple multicellular traits (*Casalone et al., 2005*; *Zara et al., 2009*), there does not appear to be a correlation among the numerous adherence phenotypes (*Hope and Dunham, 2014*). Despite overlap in conserved signaling and regulatory networks governing the traits, the ability of a strain to express multicellularity in one form does not necessarily suggest the ability to express it in another (*Cullen and Sprague, 2012*). This is not entirely surprising, since different environmental conditions likely favor different multicellular phenotypes.

Here, yeast populations were artificially selected for adherence ability in one context, in order to determine whether it led to an increase in virulence in another. Specifically, yeast were evolved to adhere to a plastic bead (*Poltak and Cooper, 2011*), then tested against wax moth larvae to estimate virulence. In replicate populations of two genetic backgrounds, the yeast increased in their ability to express multicellularity in numerous forms. Not all multicellular phenotypes responded in the same way, with pseudohyphal growth appearing to evolve independently from plastic adherence, biofilm formation, and flor formation. This phenotypic evolution demonstrates the complexity of the interacting genetic networks underlying yeast multicellularity. Along with these correlated effects of selection, the yeast also became more virulent. Our results experimentally demonstrate that selection on dual-use traits outside of a host environment can inadvertently favor pre-adaptations for virulence.

## Results

The evolution experiment was conducted in two genetic backgrounds. While most *S. cerevisiae* strains can be pathogenic against wax moth larvae if administered with a high enough inoculum (*Phadke et al., 2018*), we chose two strains isolated from clinical settings, and therefore, had a known tendency toward human pathogenicity as well. These strains were highly heterozygous, with tens of thousands of SNPs in each genome (*Magwene et al., 2011*); thus, the strains contained standing

genetic variation on which selection could act. The first strain, YJM311, was isolated from the bile tube of a patient in San Francisco in 1981 (*McCusker et al., 1994b*); its recombinant offspring vary in at least one form of filamentous growth (*Lenhart et al., 2019*). The second strain, YJM128, was isolated from the lung of a patient in Missouri in the 1980s (*Tawfik et al., 1989*). Both strains were engineered to constitutively express mCherry, then sporulated, digested, and germinated. Each pool of recombinant offspring was used to inoculate replicate populations.

From each ancestral strain, 10 replicate populations were evolved via serial transfer for 350–400 mitotic generations, half punctuated with sexual cycles every 40 generations, and two without beads as controls. YJM311 was evolved for 8 sexual cycles, while YJM128 was evolved for 9. Populations were grown in limiting medium in glass tubes in the presence of a plastic bead (*Figure 1A–B*). After growth, beads were washed, suspended in water, and sonicated to detach cells. The cell suspension was transferred to the next tube for growth (*Figure 1—figure supplement 1*). In YJM128, the sexual control failed to propagate after the first cycle. Therefore, a full complement of controls (three asexual and three sexual) were subsequently initiated and evolved in the same manner.

The number of cells attaching to the bead increased over time in the experimental populations (*Figure 1*; *Supplementary file 1c*). The data were analyzed with a linear mixed-effect model (LMM); coefficients for the treatment x cycle interaction estimate the effect a given treatment had over time on the number of cells adhering to a bead, as measured by hemocytometer counts. In both genetic backgrounds, the interaction coefficients were positive and sexual populations showed a larger effect than the asexual (YJM311, asexual*cycle = 0.054 (confidence interval ± 0.048), sexual*cycle = 0.184 (±0.048); YJM128, asexual*cycle = 0.47 (±0.242), sexual*cycle = 0.598 (±0.254)). Hence, as has been demonstrated in other evolution experiments in microorganisms (*Zeyl and Bell, 1997*; *Goddard et al., 2005*; *McDonald et al., 2016*; *Kosheleva and Desai, 2018*; *Kaltz and Bell, 2002*; *Lachapelle and Bell, 2012*), sexual populations showed increased adaptation in comparison to asexual populations.

In order to interrogate other effects of adherence selection, at the end of the experiment, ten individual clones were isolated from each population from four timepoints (for YJM311, cycles 2, 4, 6, 8, and for YJM128, cycles 1, 3, 6, 9). For each genetic background, over 400 clones, along with 20 ancestral recombinant offspring, were arrayed in a 96-well plate format for analysis of multicellular phenotypes.

## Plastic adherence ability

The panel of clones was first assayed for plastic adherence ability (*Figure 2A* with *Figure 2—figure supplements 1 and 2*). Plastic adherence was measured with a microplate reader that detected the fluorescence signal of cells remaining attached to a well in which culture was grown to saturation and gently rinsed. As expected, plastic adherence increased over time in the clones from experimental populations (YJM311: control*cycle = 0.008 (±0.056), asexual*cycle = 0.040 (±0.040), sexual*cycle = 0.100 (±0.040); YJM128: control*cycle = 0.016 (±0.076), asexual*cycle = 0.012 (±0.102); sexual*cycle = 0.187 (±0.114); *Supplementary file 1d*). Fluorescent signal could have evolved over the course of the experiment; indeed, from the beginning, YJM128 produced a brighter fluorescent signal than YJM311, suggesting the existence of genetic variants that could influence fluorescence expression. Despite the potential for noise in the measurement, the signal of increased adherence throughout the experiment was apparent in both genetic backgrounds. These clonal data support the results of the whole-population adherence measurement (*Figure 1*), in which cells attaching to a plastic bead were counted manually with a hemocytometer.

We next measured the ability of the clonal panel to express three other seemingly different multicellular phenotypes.

## Biofilm colony formation

The first multicellular phenotype was the ability to form complex colony morphology (CCM) on solid agar, which is indicative of the ability of a strain to form a differentiated biofilm colony, also known as a 'fluffy colony' (*Kuthan et al., 2003*; *Šťovíček et al., 2010*; *Šťovíček et al., 2014*; *Váchová et al., 2011*; *Maršíková et al., 2017*). This phenotype is correlated with another multicellular phenotype (*Hope and Dunham, 2014*), mat formation, which is a biofilm that forms on semi-solid agar (*Reynolds and Fink, 2001*); we therefore only assayed CCM. Morphology was scored after growth on solid, glucose-limiting medium using a scale from 1 to 5, with 1 representing no biofilm and 5 representing

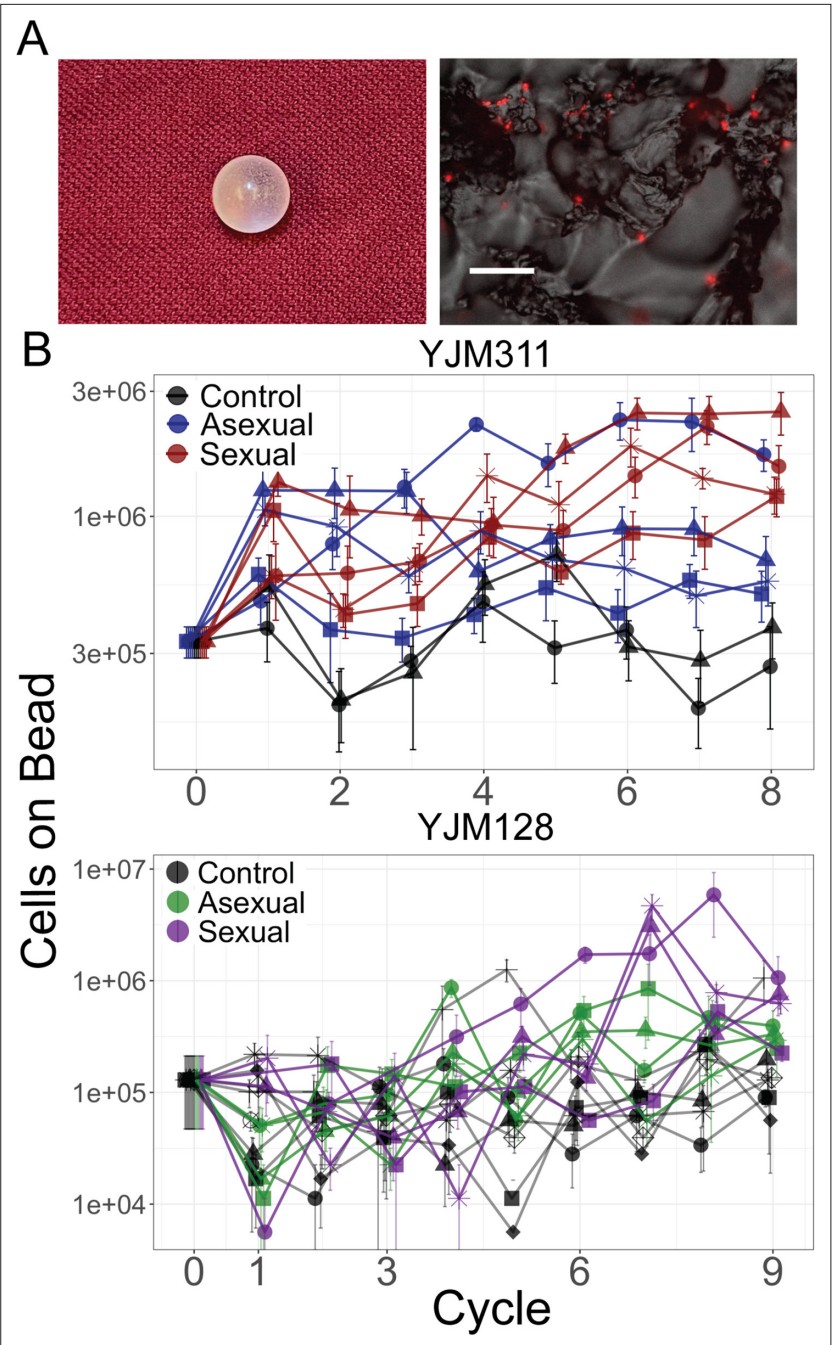

**Figure 1.** Evolution of bead adherence. (**A**) Image of a 7 mm experimental bead; close-up image with attached cells expressing mCherry, scale bar represents 50 μm. (**B**) Whole population-bead adherence of replicate populations over the experimental cycles, as estimated at the end of the experiment from cryopreserved stocks. All replicates of one background were grown and measured in one assay to test their adherence ability (including control populations that were not evolved in the presence of a bead). Y-axis plots the number of cells adhering to a plastic bead on a log scale, as estimated by hemocytometer counts (± S.E.M.). Along with the ancestral timepoint, for each population at each timepoint, cells from eight replicate beads were counted in YJM311 (670 beads in total); for YJM128, four replicate beads were counted (542 beads in total).

The online version of this article includes the following source data and figure supplement(s) for figure 1:

**Source data 1.** Counts of cells that adhered to plastic beads grown with replicate populations at different timepoints from YJM311/HMY7-derived populations.

**Source data 2.** Counts of cells that adhered to plastic beads grown with replicate populations at different

*Figure 1 continued on next page*

*Figure 1 continued*

timepoints from YJM128/HMY355-derived populations.

**Figure supplement 1.** Schematic of the Experimental Cycle.

the most structured colonies (*Hope and Dunham, 2014*; *Granek and Magwene, 2010*; *Figure 2B* with *Figure 2—figure supplements 1 and 2*).

In both genetic backgrounds, the selected populations increased in their ability to exhibit CCM compared to the ancestor, while the control populations either maintained or decreased their expression (YJM311: control*cycle = −0.024 (±0.056), asexual*cycle = 0.119 (±0.039), sexual*cycle = 0.136 (±0.039); YJM128: control*cycle = 0.007 (±0.020), asexual*cycle = 0.048 (±0.026); sexual*cycle = 0.052 (±0.030); *Supplementary file 1e* ). YJM311 evolved to exhibit stronger CCM than YJM128, despite the latter evolving for one more cycle.

## Flor formation

The second multicellular phenotype was the ability to form a flor (or velum), which is a floating mat containing cells attached to one another in an extracellular matrix (*Zara et al., 2009*). Flors form at

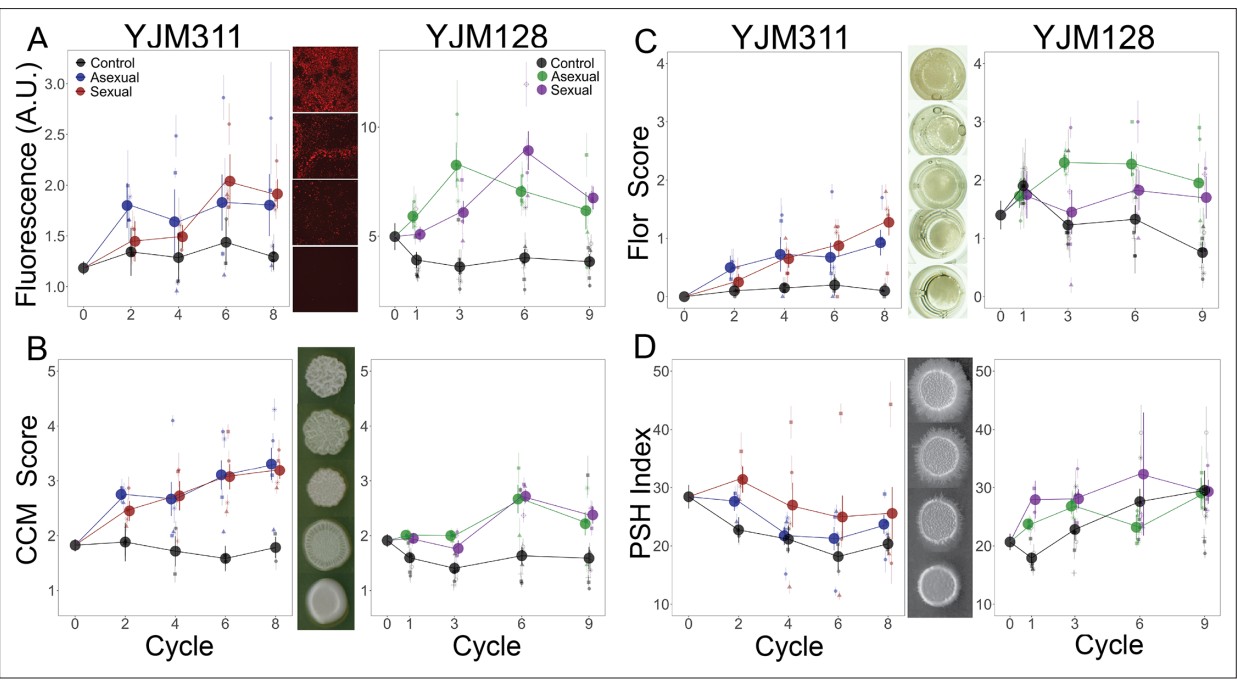

**Figure 2.** Evolution of multicellular phenotypes. Ten clones were isolated from each population at four timepoints and assayed in triplicate (except for flor formation, which had a single replicate). In all panels, large points represent the average of a treatment (asexual, sexual, control) ± 2 s.e.m.; smaller points represent the average of a replicate population ± s.e.m. Data at cycle 0 represent the average of 20 ancestral segregants. Representative images demonstrate the variation found in the phenotypes. (**A**) Plastic adherence was estimated by measuring the fluorescent signal of cells that adhered to the bottom of of a black, clear-bottom 96-well plate. (**B**) CCM was scored after growth on solid, glucose-limiting medium using the scale on the right, with 1 representing no biofilm and 5 the most structured colonies. (**C**) Flor formation was scored after growth in minimal medium using the scale on the right, with 0 representing no floating cells and 4 representing a full mat. (**D**) PSH was scored after growth on solid nitrogen-limiting medium. Images were processed to determine the percentage of growth pixels that were pseudohyphal compared to the central colony. The trajectory of replicate populations from each ancestral background can be found in *Figure 2—figure supplements 1 and 2*.

The online version of this article includes the following source data and figure supplement(s) for figure 2:

**Source data 1.** Plastic adherence, biofilm, flor, and PSH measurements for ancestor and evolved clones from the YJM311/HMY7-derived replicate populations.

**Source data 2.** Plastic adherence, biofilm, flor, and PSH measurements for ancestor and evolved clones from the YJM/128HMY355-derived replicate populations.

**Figure supplement 1.** Evolved phenotypes by replicate population for the YJM311 background.

**Figure supplement 2.** Evolved phenotypes by replicate population for the YJM128 background.

the liquid-air interface in static conditions and are most commonly found during sherry and wine making processes (*Legras et al., 2016*). Flor formation was scored after growth in minimal medium, with 0 representing no floating cells and 4 representing a full mat. The ability to form flors increased in both genetic backgrounds (*Figure 2C* with supplements 1 and 2), despite the cultures being grown with agitation. In YJM311, the ancestral clones showed no ability to generate flors, yet, its evolved populations did (control*cycle = 0.007 (±0.048), asexual*cycle = 0.075 (±0.034), sexual*cycle = 0.163 (±0.034); *Supplementary file 1f*). In YJM128, ancestral clones showed limited ability to form flors. The experimental populations either remained or increased in flor-forming ability, while control populations decreased in theirs (control*cycle = −0.109 (±0.032), asexual*cycle = 0.031 (±0.043); sexual*cycle = −0.003 (±0.048); *Supplementary file 1f*).

## Pseudohyphal growth

The final phenotype, pseudohyphal growth (PSH), is a form of filamentous growth thought to represent a foraging strategy. It is characterized by substrate invasion and incomplete separation of mother-daughter cells growing in an elongated, unipolar budding pattern (*Gimeno et al., 1992*). This phenotype is sometimes correlated with invasive growth, so only PSH was assayed. Filamentous and invasive growth have been associated with pathogenicity and virulence in *S. cerevisiae* (*McCusker et al., 1994a*; *Phadke et al., 2018*; *Palecek et al., 2002*), as well as in other fungal pathogens of humans and plants (*Lengeler et al., 2000*). PSH was scored on solid nitrogen-limiting medium; images were processed to determine the percentage of growth that was pseudohyphal compared to the central colony. Unlike the previously assayed phenotypes, the two genetic backgrounds did not evolve similarly with respect to PSH.

In YJM311, the experimental populations did not increase in their PSH ability compared to the ancestor (*Figure 2D* with supplements 1 and 2); rather, all treatments showed some loss. Throughout the cycles, a moderate level of PSH was maintained in some of the experimental populations, with one sexual replicate doubling its PSH index, but it was lost in the controls and the other experimental populations (control*cycle = −0.61 (±0.60), asexual*cycle = −0.65 (±0.42), sexual*cycle = −0.97 (±0.43); *Supplementary file 1g*).

In YJM128, both the experimental and control populations increased in their PSH ability (control*cycle = 1.29 (±0.39), asexual*cycle = 0.61 (±0.51), sexual*cycle = 0.91 (±0.55); *Supplementary file 1g*). This was true for the control population that was evolved in concert with all experimental populations, as well as the 6 control populations that were initiated subsequently.

The results from both genetic backgrounds suggest that the evolution of PSH was a response to selection in nutrient limiting conditions and not a response to selection for adherence, as the controls and experimental populations behaved similarly. Of all the adherence and multicellular phenotypes investigated, most of which appeared to increase throughout the experiment, PSH appeared to be independent from the others.

Overall, our phenotyping data show that selection on the ability to adhere to a plastic surface generated a correlated response in multiple multicellular phenotypes, and nutrient limiting conditions favored a further multicellular phenotype in one of the backgrounds.

## Hyper-multicellularity

To understand the phenotypic landscape of the evolved populations and to determine whether the different forms of multicellularity evolved in concert in individual clones, the clonal phenotype data were combined in a principal components analysis (PCA) (*Figure 3* with supplements 1 and 2). In YJM311, the loadings of the first two components, which explain 78% of the variation, show that evolved clones with the most extreme values of plastic adherence and flor formation do not tend to also excel at PSH. There were clones, however, that evolved to excel in all of the phenotypes, while not obtaining the most extreme values of the individual traits. In YJM128, the first two loadings explain 70% of the variation, and again, PSH appeared separated from the other multicellular phenotypes. Individual correlations between traits bear out this interpretation (*Figure 3—figure supplements 3 and 4*). When grouped by experimental treatments, clones from control, asexual, and sexual populations tended to occupy their own, somewhat overlapping, phenotypic space.

In both backgrounds, as the populations evolved, there were individual clones that increased in all abilities, and became 'hyper-multicellular'. Thus, a simple process of selection for plastic adherence

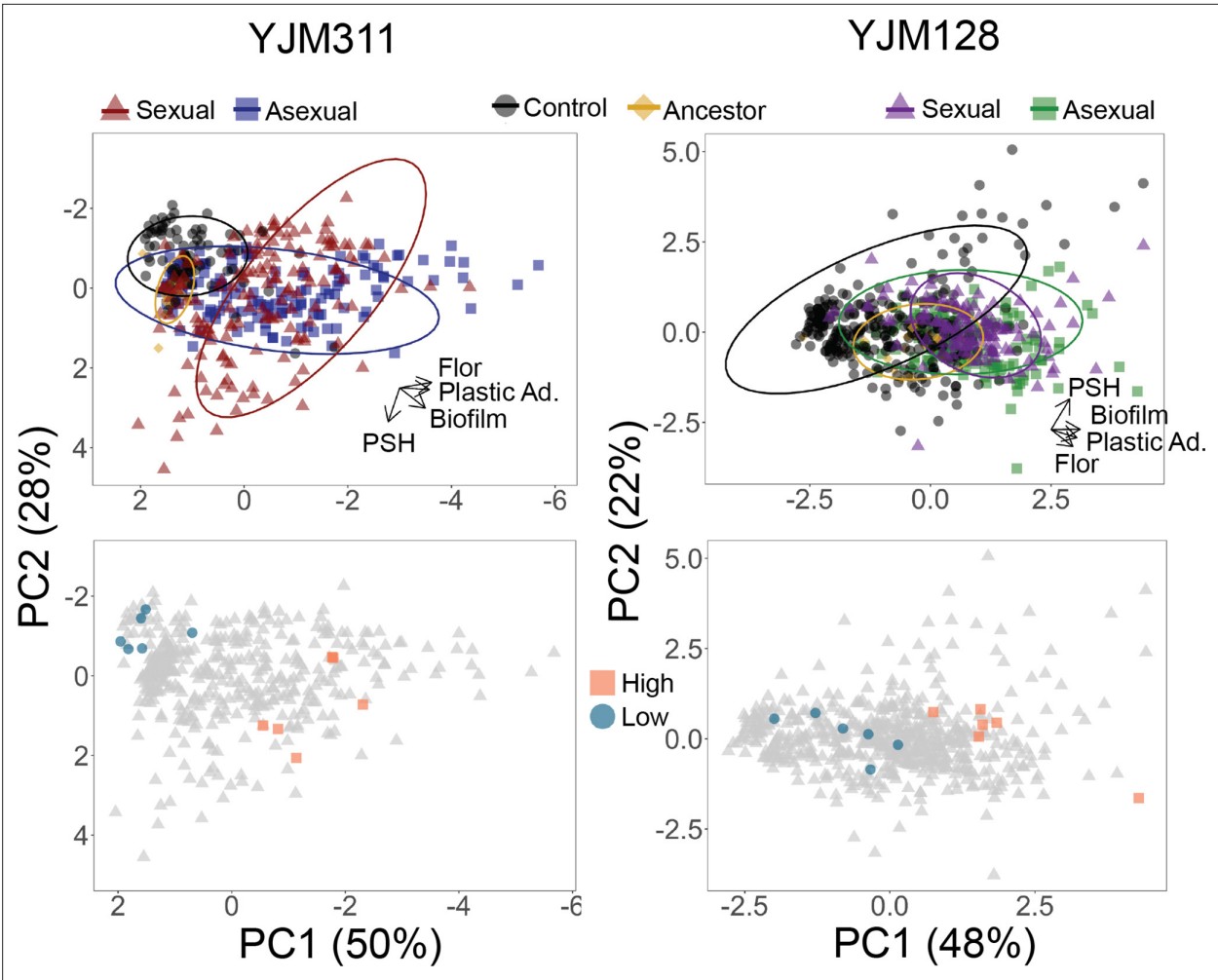

**Figure 3.** Evolved multicellularity. (**Top panel**) Principal components analysis of clones from ancestral and evolved populations. The loadings of PC1 for YJM311 were –0.616*Flor - 0.573*PA - 0.493*CCM +0.221*PSH; for PC2, they were 0.851*PSH +0.506*CCM - 0.137*Flor. In YJM128, the loadings of PC1 were 0.601*CCM +0.535*PA +0.451*Flor +0.386*PSH; for PC2, they were 0.852*PSH - 0.226*PA - 0.472*Flor. PCA with population and cycle information can be found in *Figure 3—figure supplements 1 and 2*, while individual correlations can be found in *Figure 3—figure supplements 3 and 4*. (**Bottom panel**) Principal components analysis with highlighted points representing strains chosen for virulence assays: blue circles represent low multicellularity clones; orange squares represent hyper-multicellular clones; gray triangles represent the rest of the clonal panel. In YJM311, the non-multicellular clones were chosen from ancestral and control populations, while in YJM128, they were chosen from ancestral and early experimental populations.

The online version of this article includes the following source data and figure supplement(s) for figure 3:

**Source data 1.** Survival data for wax moth larvae injected with strains from YJM311/HMY7-derived populations.

**Source data 2.** Survival data for wax moth larvae injected with strains from YJM128/HMY355-derived populations.

**Figure supplement 1.** PCA of YJM311 Evolved Populations.

**Figure supplement 2.** PCA of YJM128 evolved populations.

**Figure supplement 3.** Correlations among phenotypes for individual clones from YJM311 populations.

**Figure supplement 4.** Correlations among phenotypes for individual clones from YJM128 populations.

led to correlated effects in multiple multicellular traits. These correlated effects were apparent both at the population-level, with mean phenotypes increasing in populations over the generations, but also at the individual-level with the evolution of hyper-multicellularity.

## FLO11 length variation

One possible explanation for the increase in multiple forms of multicellularity is a change in a genetic element common to all four phenotypes. A genome-wide investigation into the genetic basis of three

multicellular phenotypes (biofilm formation, PSH, and invasive growth) in a lab strain of *S. cerevisiae* found that each phenotype appeared to have its own set of hundreds of genes underlying its expression, but also some overlap in select transcription factors and signaling pathways (*Ryan et al., 2012*). Notably, the one element that all of the traits had in common, as do other aggregative phenotypes, is the requirement of the cell adhesin, Flo11p (*Reynolds and Fink, 2001*; *Zara et al., 2005*; *Lo and Dranginis, 1998*), which allows yeast cells to adhere to surfaces and other cells (*Dranginis et al., 2007*).

Flo11p is a cell surface protein with three domains: a C-terminal that facilitates attachment to the cell wall, an exposed N-terminal immunoglobulin-like domain that mediates cell adhesion (*Kraushaar et al., 2015*), and a low-complexity, serine-threonine rich B-domain of variable length that extends the adhesion domain away from the cell (*Dranginis et al., 2007*). The tandem repeats in the B-domain have been shown to be unstable (*Fidalgo et al., 2006*; *Fidalgo et al., 2008*) and to vary in length naturally (*Zara et al., 2009*; *Oppler et al., 2019*; *David-Vaizant and Alexandre, 2018*; *Verstrepen et al., 2005*). Differences in the length of this repetitive region have been shown to affect the strength of multicellular phenotypes in some genetic backgrounds (*Zara et al., 2009*; *Fidalgo et al., 2008*).

To determine whether *FLO11* length changed throughout the experiment, amplicons of the gene were analyzed with electrophoresis in a subset of clones from the final timepoint (*Figure 4*). In the YJM311 populations, five out of eight experimental populations ended with an approximate 1000 bp length increase in some or all clones, while none of the control clones showed an increase in length. It is unknown whether the change in length was due to independent de novo mutations or selection favoring an existing allele. The similar allelic length in multiple replicate populations favors the latter explanation. It is possible that during the generation of the starting recombinant pool, there was a mutation that was not detected in the subset of ancestral clones later chosen for analysis. In this genetic background, *FLO11* length is not correlated with the strength of plastic adherence, nor with the other three multicellular phenotypes (*Figure 4—figure supplement 1*).

In YJM128, the ancestral pool likely had two alleles separated by a few hundred basepairs. The ending asexual populations appeared to have these alleles, with one much longer allele in a clone in one replicate. The ending sexual populations contained the ancestral alleles, as well as other variants both longer and shorter. Clones from one replicate could not be amplified (Sa), suggesting the possibility of a mutation in the region where the primers anneal. Again, *FLO11* length was not correlated with the strength of plastic adherence, nor with the other three multicellular phenotypes (*Figure 4—figure supplement 2*).

Thus, while *FLO11* length evolved during the experiment, it does not appear to be the cause of the correlated response to selection on adherence. However, this does not rule out the possibility that *FLO11* plays a role. It is possible that expression of the gene, through its complex regulatory network (*Rupp et al., 1999*; *Octavio et al., 2009*; *Bumgarner et al., 2009*), is related to the phenotypic response to adherence selection.

## Virulence

To test the coincidental selection-accidental virulence hypothesis, we sought to determine if the evolved changes had an effect on virulence, and were particularly interested in the unexpected evolution of hyper-multicellular clones. Virulence was measured using larvae of the greater wax moth, *Galleria mellonella*, an invertebrate model used to study microbial pathogenesis and virulence (*Pereira et al., 2018*), including in *S. cerevisiae* (*Phadke et al., 2018*). Using the phenotyping data and the PCA results as a guide, for each genetic background, we identified six hyper-multicellular clones and six non-multicellular clones (*Figure 5*, *Supplementary file 1a, b*). We attempted to identify evolved clones that excelled in all measured aggregative traits; therefore, not all were from the final time point, but all were from late in the experiment. In choosing the non-multicellular strains, clones were taken from a variety of time points, including the ancestor, control, and experimental populations at different time points. This allowed us to verify that it was the evolved hyper-multicellular phenotype and not just long-term growth in the evolution medium.

Strains were grown in the medium in which they were evolved, then washed, adjusted for density, and injected into larvae. Larval survival and pupation were monitored for the next 7 days. Different batches of larvae can be variable in their response to microbial insult; therefore, to ensure reproducibility of results, the experiment was repeated multiple times with numerous batches of larvae,

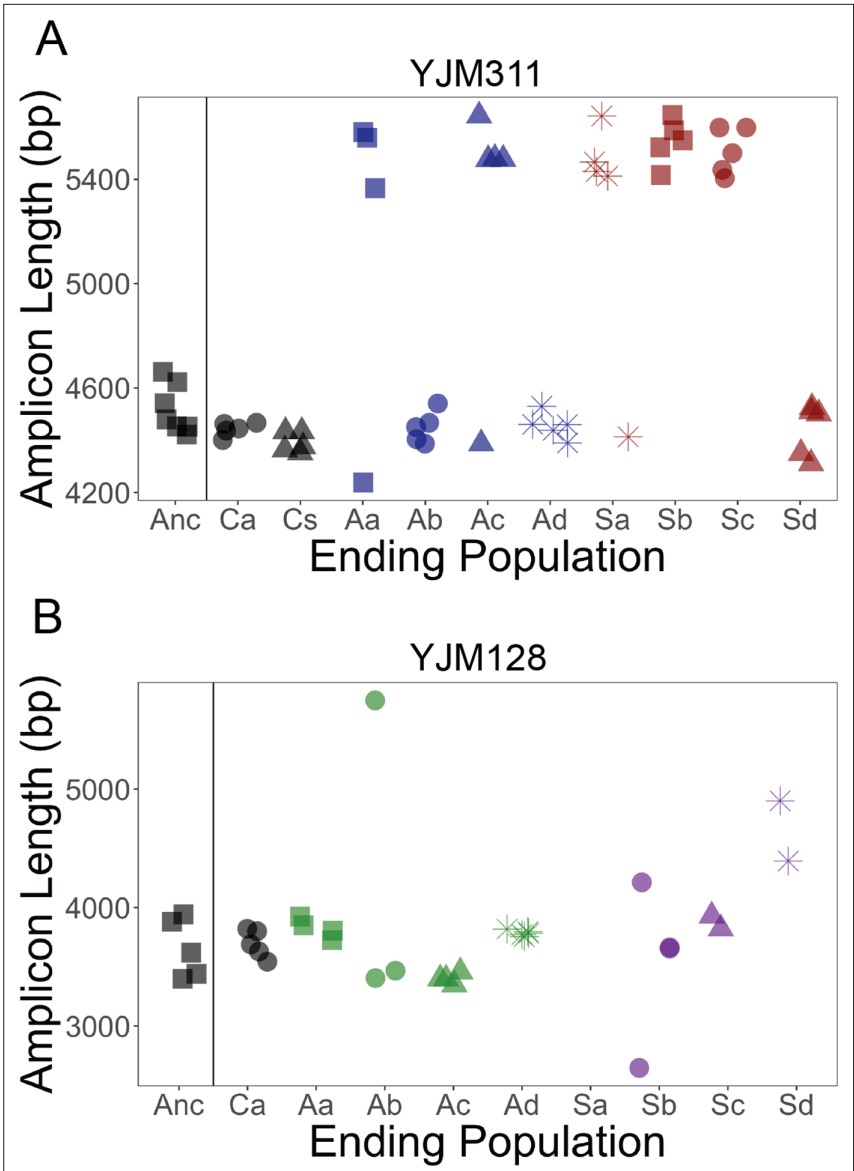

**Figure 4.** *FLO11* length evolution. In 8 ancestral clones and 5 clones per replicate population at the final cycle, the full gene was amplified and run through a BioAnalyzer to determine its length. Amplicons of this length have an accuracy of ±100 bp. x-axis: Anc refers to ancestor, A to asexual populations, and S to sexual populations, while a-d denote replicates; C refers to control populations, and a and s refer to the asexual and sexual controls, respectively. (**A**) In YJM311, it appears there were two major length alleles, with the possibility of derived variants with smaller changes in length. (**B**) In YJM128, it appears there were also two alleles, separated by ~500 bp. Clones from the final timepoint show variation in length. Correlations between length and different adherence phenotypes can be found in *Figure 4—figure supplements 1 and 2*.

The online version of this article includes the following figure supplement(s) for figure 4:

**Figure supplement 1.** *FLO11* Length by Phenotype for the YJM311 Background.

**Figure supplement 2.** *FLO11* length by phenotype for the YJM128 background.

with each batch being challenged by all strains from a genetic background. For the clones derived from YM311, of the 2400 larvae injected with yeast, 1809 did not survive through day 7. Larvae injected with a hyper-multicellular strain were 1.28 times more likely to die than those injected with a non-multicellular strain (mixed effects Cox model: coefficient = 0.249, coeff. s.e. = 0.068, p<0.001) (*Figure 5*). For the clones derived from YM128, of the 2160 larvae injected with yeast, 1113 did not survive through day 7. Larvae injected with a hyper-multicellular strain were 1.29 times more likely to

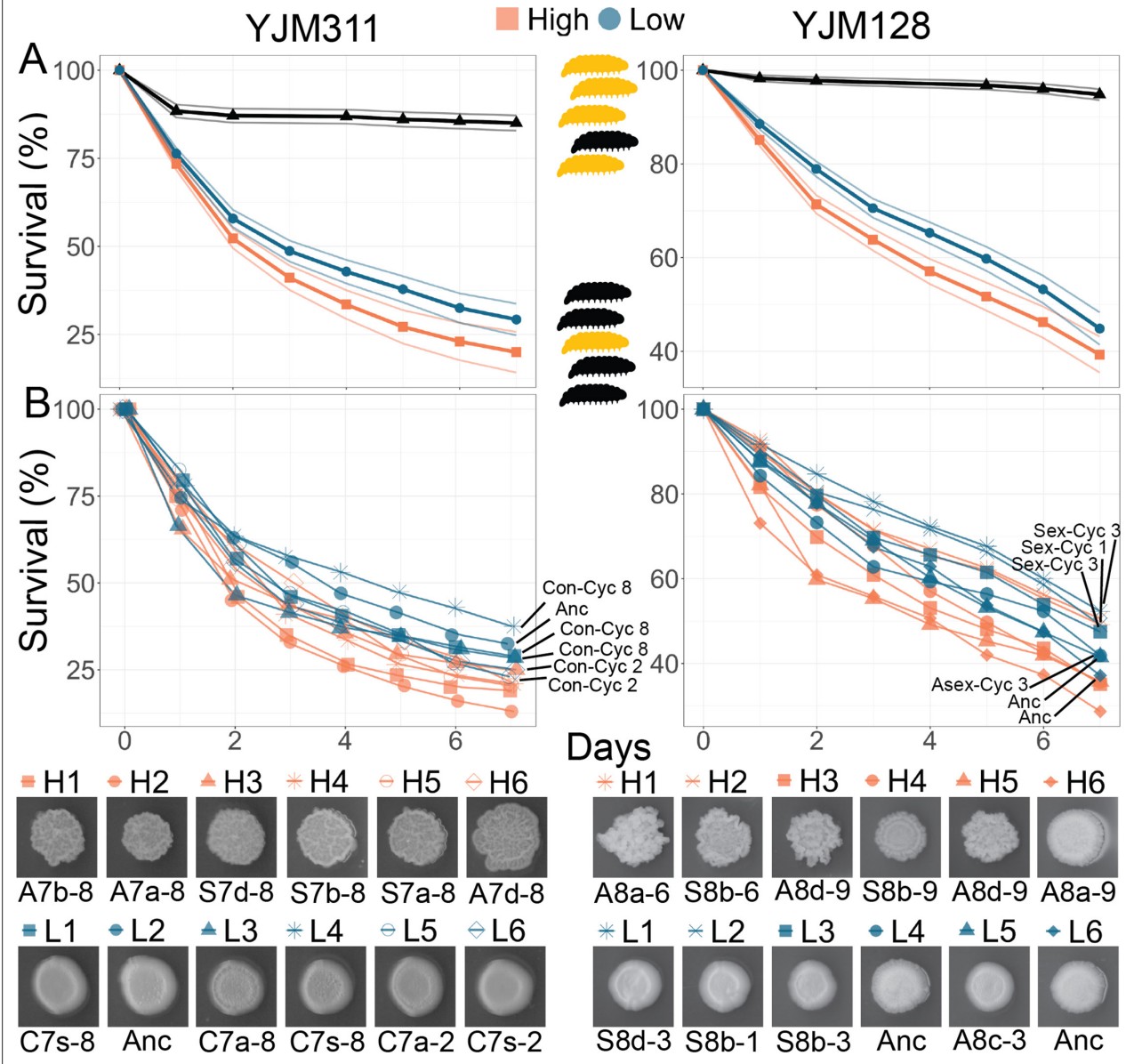

**Figure 5.** Virulence of Evolved Populations. *G. mellonella* survival curves for strains highlighted in *Figure 3*; each strain was injected into 200 larvae for YJM311-derived clones or 180 larvae for YJM128-derived clones. Points represent Kaplan-Meier estimates. (**A**) Survival curves with confidence limits for non- and hyper-multicellular treatments; black triangles represents the control treatment injected with sterile water. (**B**) Survival for individual strains along with associated CCM images. The low multicellular curves are labeled for ease of identification. Strains used in survival analyses can be found in *Supplementary file 1a and b*.

die than those injected with a non-multicellular strain (coefficient = 0.251, coeff. s.e. = 0.117, p=0.032) (*Figure 5*). When considering the survival curves of individual strains, non-multicellular strains were less virulent regardless of whether they were from ancestral, control or experimental populations.

Thus, the evolutionary changes brought about by selection for adherence to a plastic bead, led to the incidental evolution of increased virulence.

## Discussion

Our results demonstrate that selection on one yeast trait can generate a correlated response in other traits— a common feature of organismal evolution (*Price and Langen, 1992*)— but here, the correlated traits may have included those associated with virulence. In this experiment, favoring the

ability to adhere to plastic, a surface that is common in industrial, medical, and domestic settings (*Geyer et al., 2017*), led to a suite of aggregative phenotypes and increased virulence.

The coincidental selection-accidental virulence hypothesis proposes that selection for survival in harsh conditions may lead to traits that predispose microbes to virulence. However, harsh environmental conditions can also favor traits that favor the collective, or multicellular phenotypes (*Tong et al., 2022*), so it is perhaps not surprising that other forms of multicellularity increased throughout this experiment. Furthermore, in its long evolutionary history, *S. cerevisiae* has evolved the genetic capability to express multiple different multicellular phenotypes, most of which are induced in nutrient limiting conditions; the experiment presented here was performed in such conditions (glucose limited medium). Other experiments with yeast growing in nutrient limiting conditions have resulted in the unintended evolution of single aggregative behaviors (*Hope et al., 2017*). In this light, the correlated phenotypic response in this experiment is not entirely unexpected. However, the evolution of *multiple* multicellular phenotypes in *two* independent genetic backgrounds was not anticipated. Furthermore, nutrient limiting conditions may have affected PSH in our experiment, but it was not the main driver of the other multicellular phenotypes; rather, it was adherence selection that led to the evolution of hyper-multicellularity.

Previous research on a panel of environmental isolates found no correlation between the phenotypes assayed here (*Hope and Dunham, 2014*), and in a tractable lab strain capable of aggregative behaviors, each phenotype was associated with its own set of genes (*Ryan et al., 2012*). However, there is overlap in the requirement of *FLO11* and its regulators. Despite the phenotypes being induced by different nutrient signals, there are numerous conserved signaling pathways contributing to filamentous, multicellular growth of all forms (e.g. cAMP-PKA, TOR, filamentous MAPK, Rim101) (*Cullen and Sprague, 2012*; *Granek et al., 2011*). It is well known that genetic background and genetic architecture can have strong effects on the expression and correlation of traits (*Gasch et al., 2016*). In the case of the filamentous phenotypes assayed here, a genetic background that contains variants in the main signaling pathways may lead to a correlation of the phenotypes, while variants expressed later in the development of the phenotype, that are specific to a single trait, may not lead to such a correlation. The effect of the different types of genetic variants suggests that some strains and genetic backgrounds are more likely to evolve virulence from selection in the open environment.

The strains used in this experiment each contain ~50,000 heterozygous sites and differ from each other by ~25,000 SNPs. It is possible that they contained genetic variation in canonical signaling pathways, allowing for the evolution of hyper-multicellular strains. Interestingly, in both backgrounds, pseudohyphal growth appeared to evolve independently of the other phenotypes. Future research will investigate the sorting of the genetic variation, as well as the new mutations, that led to the observed phenotypic evolution in these populations.

Understanding the processes that lead to the emergence of opportunistic fungal pathogens is of increasing importance. In 2022, the World Health Organization issued its first-ever report prioritizing 19 fungal pathogens for research and public health awareness (*World Health Organization, 2022*); of these, 11 are known to live in the environment (i.e. soil, wood, etc.), including three in the highest priority group (*Cryptococcus neoformans*, *Candida auris*, *Aspergillus fumigatus*). In our experiment, it is unclear which trait was associated with increased virulence: plastic adherence, a different multicellular trait, or general hyper-multicellularity. Regardless of the specific trait causing increased virulence, the experiment demonstrated that selection for a dual-use trait in an environment that is entirely devoid of host organisms can still inadvertently lead to virulence and pathogenicity. The experiment also demonstrated the role that sex can have in increasing rates of adaptation in fungi.

Whether or not increased virulence caused by adherence selection is a general result in fungal microbes remains to be seen. In bacteria, the results are mixed. In *Burkholderia cenocepacia*, plastic bead selection led to an increase in biofilm phenotypes and mutations previously associated with chronic infections (*Poltak and Cooper, 2011*; *Traverse et al., 2013*). Yet, bead selection in *P. aeruginosa* led to a decrease in biofilm-related phenotypes. It also led to an increase in antibiotic resistance, thus mimicking changes seen in chronic infections (*Azimi et al., 2020*).

As humans continue to generate novel ecological niches at an unprecedented rate by encroaching on more habitats, using plastics unreservedly (*Iroegbu et al., 2021*), and especially, as the global climate changes (*IPCC Core Writing Team, 2023*), the potential for unintended selection grows (*Casadevall, 2020*). Clinically relevant strains of *Escherichia coli* can use microplastics in the environment as

a reservoir and can even become more virulent after recovery from the 'plastisphere' (*Ormsby et al., 2023*), and it has been shown that plastics in the environment may harbor other pathogenic taxa (*Wu et al., 2020*). Warmer water temperatures and climate disruptions have been linked to the incidence of illness caused by the marine bacterium *Vibrio vulnificus* (*Martinez-Urtaza et al., 2010*), and an increase in global temperature has been hypothesized to be related to the simultaneous emergence of *C. auris* infections on multiple continents (*Jackson et al., 2019*; *Megha et al., 2020*). More generally, the narrowing of the gap between mammalian body temperatures and the ambient environment may create opportunities for fungi to exploit new host niches (*Garcia-Solache and Casadevall, 2010*). Thus, as new selective pressures act on populations with existing abundant genetic variation, there is the opportunity to coincidentally select a new generation of accidental pathogens.

## Materials and methods

**Key resources table**

| Reagent type (species) or resource | Designation | Source or reference | Identifiers | Additional information |
|---|---|---|---|---|
| Gene (*Saccharomyces cerevisiae*) | *FLO11* | *Saccharomyces* Genome Database | YIR019C | |
| Strain, strain background (*Saccharomyces cerevisiae*) | YJM311 | Gift from Paul Magwene at Duke | | |
| Strain, strain background (*Saccharomyces cerevisiae*) | YJM128 | Gift from Paul Magwene at Duke | | |
| Genetic reagent (*Saccharomyces cerevisiae*) | HMY7 | This paper | | YJM311 homozygous for *PGK1* tagged with *mCherry-KanMX* |
| Genetic reagent (*Saccharomyces cerevisiae*) | HMY355 | This paper | | YJM128 homozygous for *PGK1* tagged with *mCherry-HygMX* |
| Biological sample (*Galleria mellonella*) | *Galleria mellonella* larvae | Vanderhorst Wholesale Inc. | | https://www.waxworms.net/ |
| Recombinant DNA reagent | pBS34 | Addgene | RRID: Addgene_83796 | Source of mCherry for tagging original strains |
| Sequence-based reagent | FLO11-for | This paper | PCR primers | GCCTCAAAA ATCCATATA CGCACACTA TG |
| Sequence-based reagent | FLO11-rev | This paper | PCR primers | TTAGAATAC AACTGGAAG AGCGAGTAG |
| Commercial assay or kit | MasterPure Yeast DNA Purification Kit | Lucigen | Cat #: MPY80200 | DNA extraction kit |
| Commercial assay or kit | Agilent DNA 7500 kit | Agilent | Cat #: 5067–1506 | PCR product length analysis kit |
| Other | 7 mm polystyrene beads | American Educational Products | Product #: 3276 | Plastic beads used in the evolution experiment (*Figure 1A*) |

### Strains

To generate strains appropriate for downstream phenotyping assays, the original diploid isolates were engineered to express a fluorescence protein by fusing mCherry to the C-terminal region of the highly expressed *PGK1* gene, generating HMY7 (YJM311 *PGK1-mCherry-KanMX*) (*Deschaine et al., 2018*) and HMY355 (YJM128 *PGK1-mCherry-HygMX*). After being subject to selection for 8–9 cycles, clones with different multicellular phenotypes were isolated from each replicate population. Original isolates were generously provided by Paul Magwene (Duke University).

### Media

Experimental populations were grown in Evolution Medium (EM; 0.17% yeast nitrogen base without ammonium sulfate and without amino acids, 0.1% glutamic acid, 0.1% dextrose) supplemented with G418 (200 µg/ml) or Hygromycin B (300 µg/ml). Cells were sporulated on solid medium (1% potassium acetate, 2% agar) and digested using an overnight zymolyase- β-glucuronidase procedure (*Goddard et al., 2005*; *Granek et al., 2013*). Phenotypes were assayed on YPD (1% yeast extract, 2% peptone, 2%

dextrose, 2% agar), low dextrose (LD) YPD (0.1% dextrose), 2 X SLAD (0.34% yeast nitrogen base without ammonium sulfate and without amino acids, 2% dextrose, 50 µmol ammonium sulfate, 2% agar), or in liquid SD (0.17% yeast nitrogen base without amino acids and with ammonium sulfate, 2% dextrose).

## Experimental evolution

HMY7 and HMY355 were grown in 10 ml YPD, sporulated, digested, grown to saturation in 10 ml EM, and used to inoculate 10 replicate populations: 4 sexual, 4 asexual, and 1 control of each reproductive type.

Experimental and control populations derived from YJM311 were evolved for 8 12-day cycles, for a total of ~350 generations; populations from YJM128 were evolved for 9 cycles, for a total of ~400 generations. In each cycle, experimental populations were grown in 10 ml of EM in a glass tube containing a sterile 7 mm polystyrene bead (American Education Products), population size ~2 x $10^8$. After 48 hr at 30 °C in a rotator drum, the bead was removed with sterile disposable forceps, washed twice, suspended in 500 µl of sterile $H_2O$ in a microcentrifuge tube, and gently sonicated (UP200St with VialTweeter, Heischler Ultrasound Technology) to detach cells from the bead. The cell suspension was used to inoculate the next 10 ml EM tube. The number of cells on the bead varied over the experiment. Control populations were also grown in 10 ml of EM in a glass tube, but without the presence of a bead. Instead, 10 µl of culture were used to inoculate the next tube, which was approximately the same number of cells as that being transferred from bead adherence in the experimental populations at the start of the experiment. After 4 serial transfers, asexual populations were refrigerated and sexual populations were sporulated for 48 hr. Asci were digested overnight, and the spores resuspended in 1 ml of EM to allow germination and mating (population size ~$10^5$ spores). Finally, the refrigerated cultures and the mated spores were used to begin the next 12-day cycle.

## Population phenotyping

To estimate adherence evolution, all populations from all cycles were assayed using the same batch of medium. 10 ml EM cultures were inoculated with cryopreserved glycerol stocks and grown for 48 hr. From these, two replicate test tubes were inoculated with two beads in each, for a total of four beads per population per time point. The cultures were grown and the beads processed as in the experimental cycle; cell counts were made using a hemocytometer with the sonicated cell suspension. This entire process was repeated a second time, for a total of 8 beads per population per cycle for YJM311 populations.

## Clonal phenotyping

Twenty clones were isolated from the ancestral population and 10 clones were isolated from each replicate population at four cycle timepoints: 2, 4, 6, 8, for YJM311, and 1, 3, 6, 9, for YJM128. The clonal strains were arrayed in a 96-well format and cryopreserved. To assay social phenotypes, saturated YPD cultures were resuspended and pinned to different media using a 96-pin multi-blot replicator (V&P Scientific no. VP408FP6), wrapped in parafilm, and incubated at 30 °C.

### Plastic adherence

Clones were grown in 200 µl EM for 48 hr in three replicate black, clear-bottom, non-treated 96-well plates. Optical density was measured, then culture was removed, and plates were gently washed with water three times and dried upside down for 1 hr. Fluorescence readings were taken with a Spectramax M2e (Molecular Devices) and used as a proxy for the number of cells that remained attached to the wells. To account for differences in growth, each fluorescence reading was divided by the optical density of the well.

### Flor formation

Clones were grown in 200 µl SD for 5 days and imaged on an Olympus SZX16 dissecting scope. Flor formation was scored using the scale in *Figure 2*.

### Complex colony morphology (CCM)

Clones were pinned to 3 replicate LD omni trays, incubated for 7 days, and imaged on an EPSON Expression 11000 XL scanner. Colonies were scored for complexity using the scale in *Figure 2*.

## Pseudohyphal growth

Clones were pinned to three replicate 2 X SLAD omni trays, incubated for 8 days, and scanned. Images were processed using a custom script that determined the percentage of colony pixels comprising the pseudohyphae (*Lenhart et al., 2019*).

## Data analysis

Bead cell count data from experimental populations were log-transformed and analyzed using a mixed effects linear model in R (*R Development Core Team, 2020*) with the lme4 package (*Bates et al., 2015*). Replicate population within treatments (control, asexual, sexual) was considered a random effect. Because all populations were begun from a single ancestral pool, the intercept was set as the mean value of the ancestor and not allowed to vary among treatments. Therefore, the only fixed effect was the interaction between cycle and treatment, which tested the differences among the slopes of the three treatments. The analysis was performed on the number of cells counted on a hemocytometer, which ranged from zero to a few hundred cells in the later cycles. Thus, the coefficients represent the effect on the number of cells per cycle on these counts. Clonal data were analyzed similarly, with the untransformed average score of a phenotype as the independent variable. Thus, the coefficients represent the effect of the treatment over time on the measurement of the phenotype. Finally, the average phenotyping data for each clone were combined for a principal components analysis in R using the *princomp* function. Figures were produced using ggplot2 (*Wickham, 2016*).

## Clones for virulence assay

The virulence assay was first conducted with clones from the YJM311 background. Clones were selected to reflect changes that occurred over the course of the entire experiment, specifically the evolution of hyper-multicellularity in experimental populations. Therefore, the low-multicellularity clones were chosen from the ancestral segregants, as well as early and ending control populations. The high-multicellularity clones were chosen from the ending experimental populations. Because CCM was correlated with plastic adherence and flor formation, it was used as an initial screen to find clones of interest (either smooth or very complex colonies); data on the other phenotypes were then investigated. Once a subset of clones was chosen, they were re-assayed to verify the phenotypes and used in virulence assays. In the second background, YJM128, clones were chosen in an attempt to isolate the effect of hyper-multicellularity, thus, other than ancestral segregants, all clones were chosen from the experimental populations using the same process. Since the high-multicellularity clones should have exhibited some level of all of the phenotypes (when possible) and PSH was evolving independently, the chosen clones do not stand out as the most extreme on the PCA plot.

## Virulence assay

10 ml EM cultures of evolved and ancestral strains *Supplementary file 1a* were grown for 48 hr, washed and resuspended in sterile water to a concentration of $10^9$ cells/ml based on hemocytometer counts. 4 µl of culture or control water was injected into the final posterior proleg of *Galleria mellonella* larvae (Vanderhorst Wholesale Inc, https://www.waxworms.net) weighing on average 180 (±20) mg using a Hamilton PB600-1 Repeating Dispenser with a 27-gauge needle. Each strain was injected into 20 larvae on the same day using the same shipment of *G. mellonella*; 20 control larvae were injected at the start of the assay and at the end. The same assay was repeated the next day with the same shipment of larvae, for a total of 40 larvae/strain/shipment. Multiple shipments were used for the virulence measurements, for a total of 200 larvae per strain for YJM311-derived strains and 180 larvae per strain for YJM128-derived strains. After injection, larvae were incubated at 30 °C and survival was monitored for 7 days; larvae that turned black and no longer responded to tactile stimulation were considered dead and removed from the population, as were larvae beginning to pupate.

Data were analyzed with a mixed effects Cox model using the coxme package (*Therneau, 2020*) in R (*R Development Core Team, 2020*). Death was recorded as the day larvae were removed from the population; larvae were censored if removed for pupation. The model included treatment (high vs. low multicellular) as a fixed effect, and strain and larval batch as random effects.

## FLO11 length

Of the clones assayed for multicellular phenotypes, 8 ancestral clones and 5 clones from the final time point of each replicate population were chosen for length analysis. Genomic DNA was extracted using the MasterPure Yeast DNA Purification Kit (Lucigen). *FLO11* was amplified with Phusion polymerase (New England BioLabs) and primers targeting the entire gene (forward: GCC TCA AAA ATC CAT ATA CGC ACA CTA TG, reverse: TTA GAA TAC AAC TGG AAG AGC GAG TAG). Cycle conditions followed manufacturers recommendations and included a melting temperature of 58 °C and 3 min extension time. Gene length was estimated by running PCR amplicons through the Agilent 2100 BioAnalyzer using the Agilent DNA 7500 kit (as in ref [*Oppler et al., 2019*]).

## Acknowledgements

We thank Paul Magwene for strains, and Joseph Heitman and Anna Averette for guidance with *G mellonella* assays. The research was funded by National Institutes of Health grant R15GM122032 and National Science Foundation grant DEB-1839555 to HAM, and William & Mary Charles Center Summer Fellowships to LIE, DM, DVM, and JAS.

## Additional information

### Funding

| Funder | Grant reference number | Author |
|---|---|---|
| National Institutes of Health | R15-GM122032 | Helen A Murphy |
| National Science Foundation | DEB-1839555 | Helen A Murphy |

The funders had no role in study design, data collection and interpretation, or the decision to submit the work for publication.

### Author contributions

Luke I Ekdahl, Data curation, Investigation, Methodology, Writing – review and editing, Conducted phenotyping assays; conducted all virulence assays; Juliana A Salcedo, Data curation, Investigation, Methodology, Evolved all the populations and conducted phenotypic assays; Matthew M Dungan, Investigation, Conducted phenotyping assays; Despina V Mason, Methodology, Developed the evolution protocol; Dulguun Myagmarsuren, Investigation, Conducted FLO11 length analysis; Helen A Murphy, Conceptualization, Data curation, Formal analysis, Funding acquisition, Writing - original draft, Project administration, Writing – review and editing

### Author ORCIDs

Helen A Murphy ⓘ http://orcid.org/0000-0002-4363-4543

### Decision letter and Author response

Decision letter https://doi.org/10.7554/eLife.81056.sa1
Author response https://doi.org/10.7554/eLife.81056.sa2

## Additional files

### Supplementary files
• MDAR checklist

• Supplementary file 1. This file contains tables that list strains used for injections and full results from linear models. (**a**) YJM311 Strains used in virulence experiments. (**b**) YJM128 Strains used in virulence experiments. (**c**) Results of the mixed-effect linear model for whole-population cell count data over experimental cycles. (**d**) Results of the mixed-effect linear model for clonal plastic adherence data. (**e**) Results of the mixed-effect linear model for clonal CCM data. (**f**) Results of the mixed-effect linear model for clonal flor data. (**g**) Results of the mixed-effect linear model for clonal

PSH data.

## Data availability

All data generated or analysed during this study are included in the manuscript and supporting file. Source data files have been provided for Figures 1, 2, and 5. Original and evolved strains are available upon request to the corresponding author.

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
