## [Editor Report]

Using experimental evolution and virulence assessment in a model system, this valuable study examines how yeast virulence can coincidentally evolve following selection for plastic adherence. The strength of evidence is solid. The work presents interesting experimental systems, and the findings will be of interest to investigators in the field of experimental evolution and evolution of fungal pathogens.

---

## [Decision Letter]

**Decision letter after peer review:**

Thank you for submitting your article "Selection on plastic adherence leads to hyper-multicellular strains and incidental virulence in the budding yeast" for consideration by *eLife*. Your article has been reviewed by 2 peer reviewers, and the evaluation has been overseen by a Reviewing Editor and Arturo Casadevall as the Senior Editor. The following individual involved in the review of your submission has agreed to reveal their identity: Daniel Smith (Reviewer #1).

Essential revisions:

The reviewers have provided detailed comments and made recommendations for improving the paper. Furthermore Reviewers 1 and 2 had discussions among themselves and with me regarding what would constitute a successful revision and here is a distillation of that discussion, which is in addition to the Reviewer comments below.

– As noted by Reviewer it is very important to address the relationship to previous work on 'coincidental selection for virulence' and place your findings in that context.

– You must address the concerns about how the virulence assays were conducted. For example, It was not clear to Reviewer 1 why the paper lacked a simple 'ancestor' versus 'evolved with beads' versus 'evolved without beads' comparison. The concern here is that pooling across generations and stratifying by a correlated phenotype could introduce biases that confound the question of phenotype under observation.

– There was concern about the pooling of the data. Although the unpooled data was in the supplemental data this should be on the main figures. Pooling is a problem because the isolates come from different cycles and conditions, and the readers should be able to see which are the ancestral/control and late-cycle experimental isolates. Showing that enhanced virulence phenotype is present in 10-12 of the hyper multicellular isolates makes their data stronger than showing them pooled as one.

– The paper needs to do a better job in describing the selection of the 20 low hyper multicellular strains and the 20 high hyper multicellular strains and include the Supplemental Tables 1 and 2 in the main text. This could help the reader see which strains were used and also show that the low hyper multicellular strains are generally control or ancestral isolates. They might also be able to write which strain is which in the virulence graph keys.

Please know that I will be sending this paper back to the reviewers if a revised version is submitted to *eLife*. Hence, I encourage you to address all the issues raised by the review. Personally, I think this is a very interesting paper and I hope that it can be revised such that it would be acceptable to the journal. The comments/criticisms are extensive but if addressed will result in a much stronger paper.

*Reviewer #1 (Recommendations for the authors):*

Overall, I found this paper very interesting, and I think the experiments were thorough and well-designed, and I am very impressed with the volume of the G. mellonella infections. It is amazing how much the yeast changed after the 8/9 cycles of selection for adhesion to plastic. I think the paper does a good job describing the multicellular (and hyper-multicellular) phenotypes that arose from the adhesion to plastic. I also found it interesting that the PSH evolved independently of plastic adhesion and was likely due to nutrient deprivation in the media. Below are some suggested edits and changes you could make to the presentation of the data that could help effectively show the massive amount of work done.

Questions:

1. How many of the bead-adhering cells get removed during the sonication and washing process? Are there strongly adherent cells remaining? If so, would you then be selecting for medium-plastic rather than strong adherence?

2. Have you looked at the cell surface hydrophobicity of some of these strains? A MATH assay (Microbial Adhesion to Hydrocarbon) is relatively straightforward to try, although it might be fickle and highly variable if the baseline hydrophobicity is low. I would expect the plastic-selected ones to have higher hydrophobicity which is sometimes associated with enhanced virulence.

3. Line 114: you mention the possibility of fluorescence evolution over time. Although it seems unlikely (and you give good reasons why), if you wanted to verify could you measure the mean fluorescence intensity of an individual cell? (i.e. take an image under the microscope and measure the mean gray value of the individual cells using an image processing software like ImageJ/FIJI. A mean gray value of 0 is black/no fluorescence signal and 255 would be white/brightest fluorescent signal. It would be expected that the fluorescence of each cycle would be equal.)

4. Do you think the phenotypes are reversible? Genetics is discussed a lot, but there is a possibility that this is epigenetics-based.

Data Presentation:

1. In my opinion, there is a lot of great data visualization that is put in the supplementary figures that could be included in the main figures/table. Namely, I think including a version of Table S1 and S2, Figure S6/7, Figure S8/9 (as part of Figure 4), and Figure S10 (instead of the current Figure 3B) in the main manuscript would be helpful.

2. Figure 1 may benefit from including Figure S1 as the first panel. I think it might help the reader better understand the evolution schema. Figure S1 could also be labelled directly on the illustration to improve understanding. Figure 1 should also have a scale bar in the fluorescence microscope mage in Panel A.

3. The presentation of the linear mixed-effect model data on lines 95-101, 112-114, 137-139, 151-152, 172-177 is unclear and took a while to understand. It is not clear what the units are for these data, which would be important to know and better understand the results. Including the units and removing the formula and including that information in the methods it would help the clarity. Could you compare these counts in any other way such as averaging the values of the sexual and asexual groups instead at each cycle and do an ANOVA comparing to the control?

4. Connecting the dots of the average values over the cycles in Figure 2 would help illustrate the trends better.

5. Can you correlate the fungal virulence with the quantified phenotypes of the individual strains (Similar to Figure S6 and S7) using either % mortality, pathogenic potential (PP or PPT), or median survival time as measures? I think that maybe correlating virulence to a specific multicellular phenotype show if one, in particular, is responsible for the increased virulence (as mentioned in Line 328-332).

6. Similarly, have you looked at if the longer FLO11 genotype correlated with the virulence?

7. If space permits, for Figure 3, write "high multicellularity" and "low multicellularity" in the key instead of "high" and "low" for better clarity.

8. In Figure 4, it might be easier to understand the X axis if instead of (a-d) it would be 1-4, and/or there was a key in the figure itself rather than just in the figure legend. I am also not sure if "Cs" is a typo or if I cannot find what the lowercase "s" stands for. I also think the data in Figure S8/9 is really interesting and would be nice to include in the main Figure 4.

General suggestions:

1. In general, I think the introduction would benefit from including more background information and elaborating on some of the points that I mention below.

2. It might be helpful to add more examples of predominately saprophytic/environmental microbes that also happen to be pathogens (lines 10,11).

3. The sentence in Line 20 about C. auris in the environment seems to be missing the part of the sentence describing the environmental pressure (salinity? Temperature?) that theoretically drove its adaptation into being a good pathogen.

4. An additional sentence or two after Line 22 discussing how adhesion, biofilm formation, and hydrophobicity are involved in virulence would be helpful (i.e. helping microbes stick to tissue and invade), and additional examples of microbes where adhesion is important like *Candida albicans*.

5. While there are reports of *S. cerevisiae* infection, they are relatively rare and the fungus is better known as a very useful model for fungal biology in the lab. Elaborating on the environmental and research role of *S. cerevisiae* would be good in the introductory paragraph (beginning line 28).

6. Is there a specific reason why the strains are referred to as YJM311 and YJM128 in the results and discussion, but in the methods are referred to as HMY7 and HMY355 (the strain named after being engineered to be drug-resistant and express mCherry)?

7. Since the virulence data is in Figure 3, it should be brought up in the results before Figure 4 (FLO11 length), or the figures can be combined/rearranged accordingly.

8. I think you are underselling your results in your description lines 299-305 because, besides PSH, the control conditions did not show increased multicellular phenotypes in the same way that the selected cultures did.

9. Might help to put a standalone sentence about the dramatic increase in plastic/microplastic, hydrocarbon, and metal pollution in the environment following the sentence from 334-336.

10. Number and unit should be separated by a space, for example, 10ml should be 10 ml, 200µl should be 200 µl, etc. Mostly in the methods section text.

*Reviewer #2 (Recommendations for the authors):*

I managed to misread 'plastic evolution' several times, attempting to interpret it as 'evolution of plasticity' (a topic that does intersect with virulence, see eg ref [1]). To avoid others making a similar mistake, perhaps rephrase it as 'evolution on plastic' or similar?

The FLO gene length analysis left me wondering why not look at sequence or structure? Looking only at gene length seemed a bit of a blunt instrument to get at adhesion behavior, and the lack of association doesn't tell us much (to be fair, this point is made by the authors).

Turning to analysis choices, the authors focus on analyzing changes in time, which is entirely reasonable (e.g. 'does bead attachment increase/decrease thru evolutionary time'?). But this leaves open whether the evolutionary trajectories end up in significantly different places – so I would suggest analyzing differences across endpoints (eg for evolved with beads versus evolved without beads).

[Editors' note: further revisions were suggested prior to acceptance, as described below.]

Thank you for resubmitting your work entitled "Selection on plastic adherence leads to hyper-multicellular strains and incidental virulence in the budding yeast" for further consideration by *eLife*. Your revised article has been evaluated by a Senior Editor and a Reviewing Editor.

The manuscript has been improved but there are some remaining issues that need to be addressed, as outlined below:

*Reviewer #1 (Recommendations for the authors):*

Thank you to the authors for revising the manuscript and including as many of the edits and changes that were feasible, especially given the circumstances of life. I think the revised manuscript is in good shape, and I don't think any of the not performed suggested experiments were necessary for the manuscript or for the final outcome of the project. One strong suggestion is that the Figure 1 supplemental with the graphical abstract can be better labelled directly on the figure so there are labels and descriptions without having to read the figure legend.

*Reviewer #2 (Recommendations for the authors):*

In this revised manuscript, the authors have addressed several of my prior concerns, leading to clear improvements. Overall the paper is generally well written, and I appreciate the thoughtful and frank rebuttals. Yet the responses to issues 2-4 (especially 3) still leave me with ongoing concerns.

1. Integration with prior literature on 'co-incidental selection'.

The authors do a very nice job in connecting the literature threads on 'co-incidental selection' and 'accidental virulence'. This will help the field.

2. Experimental evolution controls.

The authors clarify that Figure 1 bead attachment data is from a separate phenotyping experiment under standard bead conditions, run after the experimental evolution was completed. This clarifies my primary concern. I suggest the authors add a sentence to spell this out in the figure legend, to avoid similar confusion. I also note that the specific design of the control experiment is not specified in the methods. Given there was no bead, what exactly was the passaging protocol? (I expect it was some standard volume transfer). How comparable is this design to the bead design, in terms of approx. number of cells transferred? Substantial differences in the passaging bottleneck could raise issues in the interpretations of comparisons to the control.

3. Sample pooling for virulence assays.

The authors provide an honest summary of their rationale for their experimental design, but little in the way of changes to the MS. As things stand, I still have significant concerns over the current MS. The specific rationale for individual strain selection into their 'low/high' groups is quite mysterious, which further reduces confidence. Line 344: 'using PCA results as a guide,..' – what does this mean? The selected strains are not close to extreme positions in the PCA (see PCA lower panels). At a minimum, the authors need to present clear and unambiguous criteria for strain selection. Without this, I am concerned that the contrasts don't even meet the authors goal of directly filtering on high versus low multicellularity phenotypes. Even with a more specific, unambiguous set of criteria for strain selection, my previous concerns remain. By selecting the virulence assay on phenotypes (seemingly blind to passage time or treatment), we weaken the ability to directly assess the impact of different experimental treatments on evolutionary outcomes.

4. FLO gene length typing is a blunt instrument.

The authors agree, they have sequence data but are holding it back for another paper. The current data provides very little insight into the molecular mechanisms underlying the phenotypic differences reported.

In summary, the MS is improved, and it provides useful data that will benefit the field. Yet my earlier concerns are not fully addressed. Addressing concerns 2-4 would help increase the rigor and repeatability of the research.

---

## [Author Response]

Essential revisions:The reviewers have provided detailed comments and made recommendations for improving the paper. Furthermore Reviewers 1 and 2 had discussions among themselves and with me regarding what would constitute a successful revision and here is a distillation of that discussion, which is in addition to the Reviewer comments below.– As noted by Reviewer it is very important to address the relationship to previous work on 'coincidental selection for virulence' and place your findings in that context.

We wholeheartedly agree with this criticism. Our previous literature search and review focused exclusively on the fungal literature, and was missing this important context. We had no intention of ignoring previous work and apologize for the omission. To the best of our ability, given the space constraints, we have updated the Introduction to include this hypothesis and the subsequent work testing it. We also expanded this section to include more review of the fungal literature, as it is comparable to that which we highlight from the bacterial literature. This review of the literature has even generated new ideas for our own research, and we are grateful for the urging of the reviewers.

– You must address the concerns about how the virulence assays were conducted. For example, It was not clear to Reviewer 1 why the paper lacked a simple 'ancestor' versus 'evolved with beads' versus 'evolved without beads' comparison. The concern here is that pooling across generations and stratifying by a correlated phenotype could introduce biases that confound the question of phenotype under observation.

This is a reasonable criticism and to be honest, one we struggled with when we were choosing the strains to assay for virulence. As noted in the paper, the evolution of hyper-multicellular strains was unexpected and surprising, and we were therefore focused on the effect of this trait (or suite of traits) on virulence. Hyper-multicellularity was only observed in populations evolved with a bead, and not in the ancestral clones or those evolved without a bead.

For the first genetic background that we assayed (YJM311), for the low-multicellular treatment, we exclusively chose ancestors and clones from populations evolved without beads (cycle 2 and cycle 8). The results from this design (or strain choice) would allow us to say that the virulence increased in populations evolved with beads (and was not due to evolution in the medium alone). When it came time to choose strains/clones from the second background (YJM128), for the low-multicellular treatment, we again chose a few ancestors, but were also interested in finding low-multicellular clones from populations evolved with a bead. This would allow us to say that clones evolved in the same conditions, but with different resulting multicellular phenotypes, had different patterns of virulence. As can be seen by the clones we chose, most of those strains came from earlier in the experiment (cycles 1 and 3). For the same reason, we also chose hyper-multicellular strains from a timepoint earlier than the last one (cycle 6). We were really trying to get at whether it was the hyper-multicellularity or evolution with a bead, but were limited to what phenotypes evolved at different timepoints in the experiment.

Given the number of larvae required to assay the virulence of each clone, we could not reasonably assay more strains than we did. For better or worse, we made the choices described above. In our first genetic background, we could in principle make the “simple” comparison of ancestor vs. evolved with bead, and ancestor vs. evolved without bead, but the number of strains in the different categories would be unequal. Instead, we choose to focus on the phenotype, which was what we were interested in.

To be completely transparent and for ease of understanding, we have labeled all the low multicellular survival curves in the figure (Figure 5), which should help the readers navigate the results.

– There was concern about the pooling of the data. Although the unpooled data was in the supplemental data this should be on the main figures. Pooling is a problem because the isolates come from different cycles and conditions, and the readers should be able to see which are the ancestral/control and late-cycle experimental isolates. Showing that enhanced virulence phenotype is present in 10-12 of the hyper multicellular isolates makes their data stronger than showing them pooled as one.

As noted above, we have updated the figures to include the unpooled data. We have clearly labeled the individual curves and included images of the clonal phenotypes.

– The paper needs to do a better job in describing the selection of the 20 low hyper multicellular strains and the 20 high hyper multicellular strains and include the Supplemental Tables 1 and 2 in the main text. This could help the reader see which strains were used and also show that the low hyper multicellular strains are generally control or ancestral isolates. They might also be able to write which strain is which in the virulence graph keys.

We have done our best to comply with this request (see above).

Please know that I will be sending this paper back to the reviewers if a revised version is submitted to eLife. Hence, I encourage you to address all the issues raised by the review. Personally, I think this is a very interesting paper and I hope that it can be revised such that it would be acceptable to the journal. The comments/criticisms are extensive but if addressed will result in a much stronger paper.

We appreciate the encouragement and hope the revised version is acceptable.

Below we respond to the suggestions from each reviewer, focusing on the ones that were not included in the summary above.

Reviewer #1 (Recommendations for the authors):Overall, I found this paper very interesting, and I think the experiments were thorough and well-designed, and I am very impressed with the volume of the G. mellonella infections. It is amazing how much the yeast changed after the 8/9 cycles of selection for adhesion to plastic. I think the paper does a good job describing the multicellular (and hyper-multicellular) phenotypes that arose from the adhesion to plastic. I also found it interesting that the PSH evolved independently of plastic adhesion and was likely due to nutrient deprivation in the media. Below are some suggested edits and changes you could make to the presentation of the data that could help effectively show the massive amount of work done.Questions:1. How many of the bead-adhering cells get removed during the sonication and washing process? Are there strongly adherent cells remaining? If so, would you then be selecting for medium-plastic rather than strong adherence?

We haven’t investigated this systematically. We have occasionally looked at the beads under the microscope to verify that the cells have been removed. It is certainly possible that after long-term selection, there could be cells that adhere so strongly that sonication does not remove them and they would not be included in the inoculum for the next generation. At this point, we have no evidence for that occurring.

2. Have you looked at the cell surface hydrophobicity of some of these strains? A MATH assay (Microbial Adhesion to Hydrocarbon) is relatively straightforward to try, although it might be fickle and highly variable if the baseline hydrophobicity is low. I would expect the plastic-selected ones to have higher hydrophobicity which is sometimes associated with enhanced virulence.

This is an interesting suggestion and one we may use for future research, as we are very interesting in the evolved multicellular changes. But in our opinion, this is out of the scope of the current manuscript, as it doesn’t directly address the idea of virulence evolution.

3. Line 114: you mention the possibility of fluorescence evolution over time. Although it seems unlikely (and you give good reasons why), if you wanted to verify could you measure the mean fluorescence intensity of an individual cell? (i.e. take an image under the microscope and measure the mean gray value of the individual cells using an image processing software like ImageJ/FIJI. A mean gray value of 0 is black/no fluorescence signal and 255 would be white/brightest fluorescent signal. It would be expected that the fluorescence of each cycle would be equal.)

This is also an interesting suggestion, but since the plastic adherence assays show the expected patterns (increase over the cycles); thus, in our opinion, this extra assay is unlikely to contribute significantly to the manuscript.

4. Do you think the phenotypes are reversible? Genetics is discussed a lot, but there is a possibility that this is epigenetics-based.

While it is possible that there is an epigenetic component, we believe the majority of the phenotype is genetic. Given the abundance of evidence that there is a genetic basis for each of the traits assayed here (mapping studies, etc.) and the extensive amount of genetic variation in each clinical isolate (~40-50K heterozygous sites), it is reasonable to surmise that the sorting of this genetic variation is leading to the gradual evolution of increase in multicellularity throughout the experiment.

Data Presentation:1. In my opinion, there is a lot of great data visualization that is put in the supplementary figures that could be included in the main figures/table. Namely, I think including a version of Table S1 and S2, Figure S6/7, Figure S8/9 (as part of Figure 4), and Figure S10 (instead of the current Figure 3B) in the main manuscript would be helpful.2. Figure 1 may benefit from including Figure S1 as the first panel. I think it might help the reader better understand the evolution schema. Figure S1 could also be labelled directly on the illustration to improve understanding. Figure 1 should also have a scale bar in the fluorescence microscope mage in Panel A.

We thank the reviewer for the compliment and have taken the suggestion to update Figure 4 (now Figure 5) by including the data from Figure S10 and Table S1. For the other sets of supplementary figures, we intend to take advantage of the *eLife* format and link these supplementary figures to the main figures in the text. Specifically, when Figure 3 is viewed (the PCA), there will be links to Figures S6/7 to show the individual correlations. Similarly, when Figure 4 (FLO11 evolution) is viewed, there will hyperlinks to Figures S8/9. In this way, the supplementary figures will be viewed in the context in which they are discussed in the manuscript. This is also true for Figure 1 with a hyperlink to S1.

3. The presentation of the linear mixed-effect model data on lines 95-101, 112-114, 137-139, 151-152, 172-177 is unclear and took a while to understand. It is not clear what the units are for these data, which would be important to know and better understand the results. Including the units and removing the formula and including that information in the methods it would help the clarity. Could you compare these counts in any other way such as averaging the values of the sexual and asexual groups instead at each cycle and do an ANOVA comparing to the control?

We thank the reviewer for pointing out the issue of interpreting the coefficients. We have updated the methods to clarify, and added language to the results when the linear model is first introduced.

4. Connecting the dots of the average values over the cycles in Figure 2 would help illustrate the trends better.

We have taken this suggestion and made this edit.

5. Can you correlate the fungal virulence with the quantified phenotypes of the individual strains (Similar to Figure S6 and S7) using either % mortality, pathogenic potential (PP or PPT), or median survival time as measures? I think that maybe correlating virulence to a specific multicellular phenotype show if one, in particular, is responsible for the increased virulence (as mentioned in Line 328-332).6. Similarly, have you looked at if the longer FLO11 genotype correlated with the virulence?

At the reviewer’s suggestion, we have analyzed the virulence data using a Cox survival analysis and extracted the hazard ratio for each strain. Not surprisingly, each of the multicellular phenotypes was correlated with this measure of virulence. The reason is that the two categories of strains were chosen to have either all or none of the multicellular phenotypes, and as noted in the manuscript, one category had a higher virulence than the other (i.e., high and low multicellularity). Thus, we cannot point to one particular phenotype as being responsible. *FLO11* was not correlated with virulence.

7. If space permits, for Figure 3, write "high multicellularity" and "low multicellularity" in the key instead of "high" and "low" for better clarity.

The updated figure does not have the space, but the figure legend clarifies the groups.

8. In Figure 4, it might be easier to understand the X axis if instead of (a-d) it would be 1-4, and/or there was a key in the figure itself rather than just in the figure legend. I am also not sure if "Cs" is a typo or if I cannot find what the lowercase "s" stands for. I also think the data in Figure S8/9 is really interesting and would be nice to include in the main Figure 4.

We thank the reviewer for pointing this out; the legend has been updated to clarify the nomenclature. As noted above, Figures S8-9 will be hyperlinked to the figure in the main text.

General suggestions:1. In general, I think the introduction would benefit from including more background information and elaborating on some of the points that I mention below.2. It might be helpful to add more examples of predominately saprophytic/environmental microbes that also happen to be pathogens (lines 10,11).3. The sentence in Line 20 about C. auris in the environment seems to be missing the part of the sentence describing the environmental pressure (salinity? Temperature?) that theoretically drove its adaptation into being a good pathogen.4. An additional sentence or two after Line 22 discussing how adhesion, biofilm formation, and hydrophobicity are involved in virulence would be helpful (i.e. helping microbes stick to tissue and invade), and additional examples of microbes where adhesion is important like Candida albicans.5. While there are reports of *S. cerevisiae* infection, they are relatively rare and the fungus is better known as a very useful model for fungal biology in the lab. Elaborating on the environmental and research role of S. cerevisiae would be good in the introductory paragraph (beginning line 28).6. Is there a specific reason why the strains are referred to as YJM311 and YJM128 in the results and discussion, but in the methods are referred to as HMY7 and HMY355 (the strain named after being engineered to be drug-resistant and express mCherry)?7. Since the virulence data is in Figure 3, it should be brought up in the results before Figure 4 (FLO11 length), or the figures can be combined/rearranged accordingly.8. I think you are underselling your results in your description lines 299-305 because, besides PSH, the control conditions did not show increased multicellular phenotypes in the same way that the selected cultures did.9. Might help to put a standalone sentence about the dramatic increase in plastic/microplastic, hydrocarbon, and metal pollution in the environment following the sentence from 334-336.10. Number and unit should be separated by a space, for example, 10ml should be 10 ml, 200µl should be 200 µl, etc. Mostly in the methods section text.

All of the above suggestions have been very useful and have been adopted, and the manuscript has been updated in the appropriate locations.

Reviewer #2 (Recommendations for the authors):I managed to misread 'plastic evolution' several times, attempting to interpret it as 'evolution of plasticity' (a topic that does intersect with virulence, see eg ref [1]). To avoid others making a similar mistake, perhaps rephrase it as 'evolution on plastic' or similar?

We have removed references to plastic evolution.

The FLO gene length analysis left me wondering why not look at sequence or structure? Looking only at gene length seemed a bit of a blunt instrument to get at adhesion behavior, and the lack of association doesn't tell us much (to be fair, this point is made by the authors).

The sequence analysis of these populations will be presented in a different manuscript. We agree that a length analysis is a blunt tool, but length variation is highly variable/mutates quickly and has been shown to be associated with adherence levels—it has even been suggested as a tool for quick evolution (Verstrepen et al., 2005. Nature Genetics). Thus, we investigated the evolution of length variation.

[Editors’ note: what follows is the authors’ response to the second round of review.]

The manuscript has been improved but there are some remaining issues that need to be addressed, as outlined below:Reviewer #1 (Recommendations for the authors):Thank you to the authors for revising the manuscript and including as many of the edits and changes that were feasible, especially given the circumstances of life. I think the revised manuscript is in good shape, and I don't think any of the not performed suggested experiments were necessary for the manuscript or for the final outcome of the project. One strong suggestion is that the Figure 1 supplemental with the graphical abstract can be better labelled directly on the figure so there are labels and descriptions without having to read the figure legend.

We have added text to the figure to supplement the legend. We believe it is clarifying and hope the reviewer agrees.

Reviewer #2 (Recommendations for the authors):In this revised manuscript, the authors have addressed several of my prior concerns, leading to clear improvements. Overall the paper is generally well written, and I appreciate the thoughtful and frank rebuttals. Yet the responses to issues 2-4 (especially 3) still leave me with ongoing concerns.1. Integration with prior literature on 'co-incidental selection'.The authors do a very nice job in connecting the literature threads on 'co-incidental selection' and 'accidental virulence'. This will help the field.

Thank you. As noted in our previous response, exploration of this literature has generated new research ideas for our lab, so we are grateful for being pointed to it.

2. Experimental evolution controls.The authors clarify that Figure 1 bead attachment data is from a separate phenotyping experiment under standard bead conditions, run after the experimental evolution was completed. This clarifies my primary concern. I suggest the authors add a sentence to spell this out in the figure legend, to avoid similar confusion. I also note that the specific design of the control experiment is not specified in the methods. Given there was no bead, what exactly was the passaging protocol? (I expect it was some standard volume transfer). How comparable is this design to the bead design, in terms of approx. number of cells transferred? Substantial differences in the passaging bottleneck could raise issues in the interpretations of comparisons to the control.

Based on this suggestion, we have added language that makes it explicit that the data from the experiment and the phenotyping are different (see Figure 1 legend).

As for the transfers, thank you for pointing this out. We should have included this detail in the manuscript, and have since added text to the methods (lines 420-423). “Control populations were also grown in 10 ml of EM in a glass tube, but without the presence of a bead. Instead, 10 ul of culture were used to inoculate the next tube, which was approximately the same number of cells being transferred from bead adherence in the experimental populations at the start of the experiment.”

However, to be clear, we cannot guarantee that the number of cells being transferred was comparable throughout the entirety of the experiment. As experimental populations became more adherent, more cells would have been transferred, and as control populations adapted to the medium, more would have been transferred. This is often the case in microbial evolution experiments when passage is by volume. At the start of the experiment, bottlenecks from passaging are consistent among treatments, but may diverge among replicates as evolution proceeds.

3. Sample pooling for virulence assays.The authors provide an honest summary of their rationale for their experimental design, but little in the way of changes to the MS. As things stand, I still have significant concerns over the current MS. The specific rationale for individual strain selection into their 'low/high' groups is quite mysterious, which further reduces confidence. Line 344: 'using PCA results as a guide,..' – what does this mean? The selected strains are not close to extreme positions in the PCA (see PCA lower panels). At a minimum, the authors need to present clear and unambiguous criteria for strain selection. Without this, I am concerned that the contrasts don't even meet the authors goal of directly filtering on high versus low multicellularity phenotypes. Even with a more specific, unambiguous set of criteria for strain selection, my previous concerns remain. By selecting the virulence assay on phenotypes (seemingly blind to passage time or treatment), we weaken the ability to directly assess the impact of different experimental treatments on evolutionary outcomes.

We appreciate and respect the reviewer’s concern about the clones we selected for the virulence assay. As noted in the previous response, we cannot now change the clones we selected, so we are left with only one option, which is to be as transparent as possible. To that end, we previously updated Figure 5 to include details about the clones that were chosen. We have now also added text to the Methods to provide details about our selection logic and criteria (lines 481-495). Unfortunately, there is nothing more we can do to address this concern. We leave the ultimate fate of the manuscript in the hands of the reviewers and editors.

For the sake of total transparency, below we provide details about our process from our lab notes. As best we could, we sought to find clones that had none of the phenotypes in the low strains and all of the multicellular phenotypes in the high strains, and also came from various specified timepoints. This ended up being more of an art than a science, which we describe below, in part because PSH evolved independently from the other phenotypes. As a reminder, we first assayed YJM311, so we chose high strains from the end of the experiment and low strains from the control populations and ancestors. When we next assayed YJM128, we were interested in isolating hyper-multicellularity as a trait, so chose low and high strains from experimental populations throughout (as well as ancestors).

The process was as follows:

1) In order to avoid sampling the same clonal background twice, we aimed to choose only one clone from a replicate population at a given timepoint.

2) Since we were ultimately interested in hyper-multicellularity, we focused our initial selection criteria more on the phenotypes that appeared to be a correlated response than on plastic adherence itself (evolved populations from later timepoints would not have survived the experiment if they were not adhering). We knew from the PCA results that CCM (complex colony morphology) strongly correlated with flor formation and plastic adherence, and PSH was evolving independently from the other phenotypes. We therefore used CCM for the first step in the screen, as colony morphology was a clear, visible phenotype with high replicability among replicate plates. Thus, for low strains, we sought smooth colonies and for high strains, we sought highly complex colony morphology (and when possible, seemingly different architectures).

3) Low Strains:

For the low strains, we aimed to find clones that had none of the multicellular phenotypes, or at the very least muted versions of them. For example, there were 20 ancestral clones from each genetic background; most of these had some form of PSH, CCM and plastic adherence. In YJM311, the ancestral clones had no flor formation, while in YJM128, they did. So for the ancestral clones, we chose the ones with the lowest scores on the combination of the three phenotypes that existed in our panel, otherwise, we would have had no ancestral clones.

For the other low strains, we began by screening the CCM plates for the smoothest colonies in the timepoints/treatments we were interested in. For YJM311, which was the first genetic background we conducted virulence assays for, we looked only for clones from control populations (early and late in the experiment). After finding the smoothest colonies from different replicates, we looked at the scores on PSH and flor formation. It was often not possible to have everything in the later timepoints: sometimes for the lowest CCM, there was slight PSH or flor formation (e.g., C7s-8-8). We therefore also looked for the lowest PSH scores, and then investigated the CCM and flor scores (e.g., C7a-8-6).

For YJM128, we attempted to get low strains from experimental populations. The procedure was similar: visibly scan the CCM plates for smoothness to begin the process, then investigate the other phenotypes. It turned out that low strains were not available later in the experiment.

4) High Strains

For the high strains, we aimed to get clones that had multiple, ideally all, of the multicellular phenotypes. We began by scanning the CCM plates for complex colony morphology (and when possible, different morphological architectures). We then investigated flor scores, plastic adherence and PSH of these clones. We were trying to balance finding clones with abilities in all phenotypes, not just the most extreme version of a single one.

For YJM 311, we were only interested in the last time point and were able to find clones from different replicate lines. In this genetic background, PSH decreased over the experiment, so we had to find clones with high CCM, PSH ability that was as least as strong as the average ancestral ability, and some flor formation.

In YJM 128, we were interested in obtaining clones from throughout the experiment; we followed a similar procedure. We began by screening CCM, then the other phenotypes. The one exception was (A8a-9-8), which had very high PSH and flor formation, but medium CCM.

After choosing the clones of interest, all strains were re-assayed to verify the scores they received in the high throughput assays.

To be clear, we stated that we used PCA results as a guide because these results gave us the insight of which traits were correlated, but it does not mean (nor did we mean to suggest) that we used the six clones with top scores on the first two PCs. We have updated the language in the manuscript. It is important to note that when looking at the PCA plot, PSH is antagonistic with the other multicellular phenotypes. Thus, the reasons we couldn’t just simply choose the most extreme points on the plot are: (1) these points do not necessarily exhibit all of the phenotypes of interest (because of the uncoupling of PSH and the others), (2) they may not have been from the correct treatment/timepoint, and (3) we were only selecting one clone from each replicate/timepoint and clones from a replicate/timepoint tended to cluster (see the supplemental figures for Figure 3).

We hope this explanation lends transparency to our process.

4. FLO gene length typing is a blunt instrument.The authors agree, they have sequence data but are holding it back for another paper. The current data provides very little insight into the molecular mechanisms underlying the phenotypic differences reported.

This manuscript is focused on the evolution of phenotypes and not molecular mechanisms. It is true that we have sequenced the replicate populations at multiple time points; however, there are ~50K SNPs being sorted in the sexual populations, and single clones fixing in the asexual populations (that differ by many SNPs, many of which likely have nothing to do with selection for plastic adherence). We are working through these messy data and have not uncovered an obvious candidate gene(s) responsible for the observed phenotypes. In fact, our QTL-mapping approach is yielding more fruitful genetic results for following up on potential alleles. We can assure the reviewer that if we had an exciting insight into the molecular mechanism, we would include it in the manuscript.

In summary, the MS is improved, and it provides useful data that will benefit the field. Yet my earlier concerns are not fully addressed. Addressing concerns 2-4 would help increase the rigor and repeatability of the research.

We have done our best to address concerns 2-4 and hope our responses are satisfactory.